# MSAMamba: Adapting Subquadratic Sequence Models To Long-Context DNA MSA Analysis

1st Vishrut Thoutam
*Software Engineering*
*High Technology High School*
Lincroft, US
vithoutam@ctemc.org

2nd Yair Schiff
*Kuleshov Group*
*Cornell University*
Ithaca, US
yairschiff@cs.cornell.edu

3rd Dina Ellsworth
*Biomedical Engineering*
*High Technology High School*
Lincroft, US
diellsworth@ctemc.org

*Abstract*—We introduce MSAMamba, a novel architecture designed to address the context-length limitation of existing DNA multiple sequence alignment (MSA) models. Traditional transformers struggle with the vast context lengths inherent in MSA genome data, mainly due to the quadratic complexity of self-attention at large batch sizes. MSAMamba leverages a selective scan operation along the sequence dimension and separates sequence length and MSA dimension processing to enhance efficiency while accounting for MSA-level inductive biases. This architecture enables scalable analysis of long DNA sequences, increasing the training context length of previous methods by 8x. In addition, we develop a row-sparse training method that significantly reduces the computational overhead of the selective scan operation during model training. We demonstrate that MSAMamba achieves performance on par with state-of-the-art (SOTA) transformer-based models in variant effect prediction tasks and exceeds their performance at larger context lengths. We also demonstrate that our model excels in GenomicBenchmarks tasks. Our results indicate that MSAMamba mitigates the computational challenges of long-context DNA MSA analysis and sets a new standard for scalability and efficiency in genomic modeling.

## I. INTRODUCTION

Advances in model sizes and architectures have brought about a revolution in sequence modeling capabilities. The introduction of recurrence [24], attention [2], and memory [22] have led to many performance improvements. The transformer model [46], commonly used in large language models (LLMs) [7], applies self-attention and implicit memory [12] to sequence modeling.

Transformers have shown impressive generalization capabilities in natural language processing, prompting researchers to extend the models' abilities to sequences beyond language. Transformers have been applied to protein sequences [29] and genomics data [40]. Recently, they have been used in DNA modeling [8]. However, The human genome consists of 3 billion base pairs, with gene sizes ranging from 10 thousand to 2 million base pairs [31]. These large DNA sequences are expensive to analyze using a transformer due to the quadratic nature of self-attention [26] and the model's instability across extended context windows [30]. Subquadratic models [37] are alternatives to transformers that show high performance in modeling global relationships across long DNA sequences [35], [41].

Raw DNA sequences lack explicit evolution and homology information. DNA multiple sequence alignments (MSAs) provide this information [45]. Models that operate on MSAs show advances in mutation detection and sequence analysis tasks [43].

However, current DNA MSA modeling architectures are not scalable to long sequences. Axial attention-based transformers [21] are the current state-of-the-art for DNA variant effect prediction using MSAs. Previous methods use this training method because it is more efficient than full self attention across the entire MSA sequence. However, the computational complexity of axial attention scales quadratically in both the sequence length and MSA dimensions (see proof III-B2). Because of this limitation, previous methods only train on sequences of up to 128 base pairs [4].

A less complex algorithm is required to support long context lengths and robust MSA sizes in DNA analysis. We introduce MSAMamba: a variant of the MSA Transformer model that replaces axial attention with a horizontal SSM selective scan [16] and a vertical attention block. This algorithm scales linearly with sequence length, allowing for more efficient model inference. We also introduce a row-level masking methodology to improve training efficiency on sparse MSA sequences. These changes decrease the computational complexity of training and fine-tuning at large context lengths, allowing us to train on longer sequences efficiently. We find that an SSM-based DNA MSA model performs similarly to SOTA transformer-based MSA models in variant effect prediction at short context lengths (128) and exceeds transformer models when training on longer sequences (1024). Additionally, MSAMamba shows improved performance in 2 out of 8 GenomicBenchmarks tasks compared to single-sequence models and transformer-based MSA models (see Table V-E).

## II. BACKGROUND

This section provides an overview of biological and AI representation concepts used to construct the MSAMamba model.

### A. DNA Terminology

Deoxyribonucleic acid is a polymer made up of 4 base nucleotides (adenine, cytosine, guanine, and thymine). The

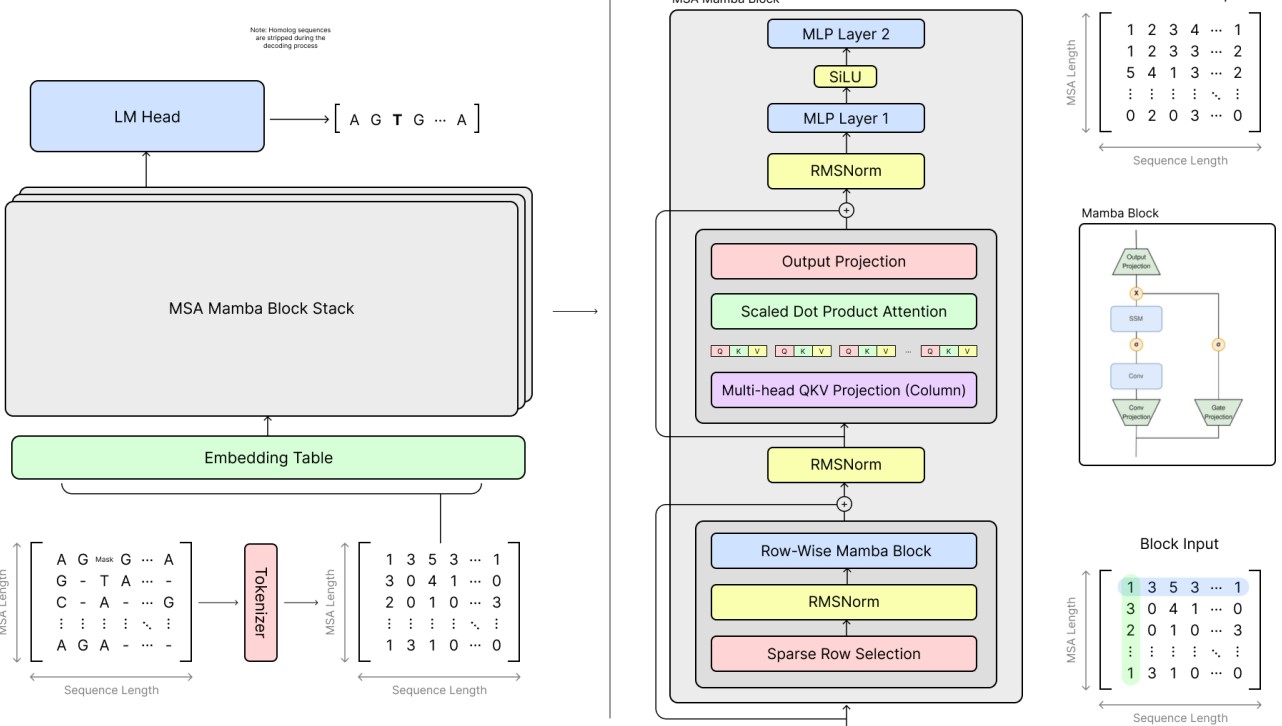

Fig. 1. A diagram of the proposed MSAMamba architecture. The architecture consists of multiple MSAMamba blocks, each containing a mamba block that acts on the sequence length dimension and masked self-attention that acts on the MSA dimension. An MLP block follows the two processes [12].

polymer forms a double helix structure from two complementary strands. DNA contains regions known as genes, which can code for different proteins to cause cellular change. Genes also consist of control sequences. These include enhancers, which can increase the DNA transcription of a specific gene into a protein; promoters, which allow the initiation of transcription; and silencers, which prevent transcription from occurring. [6]

DNA sequences also contain introns and exons. Exons contain DNA information used to form the final protein, while introns are non-coding regions that can be spliced out in different combinations to create varying gene outputs. Genes can vary in length from thousands to millions of base pairs, increasing the need for models with a large and effective context window.

**DNA MSAs** DNA Multiple Sequence Alignments (MSAs) are combinations of DNA sequences across different species. These sequences are aligned such that base pairs that evolve similarly are in the same column across genomes. Aligned columns in the MSA provide crucial evolutionary information between species. A DNA sequence for a species can be considered as a function of a different species' genome. This function consists of multiple mutations, such as insertions, deletions, and replacements. By aligning these sequences using MSA creation algorithms, models can extract evolution, conservation, coevolution, and homology information. DNA MSAs are also used to find motifs (short, repetitive sequences across genes). Implicit detection of these motifs in AI models can provide enhanced information for genome analysis. [45]

### B. Subquadratic Sequence Models

Recently, variants of state space models (SSMs) have been applied to discrete sequence modeling and have shown impressive results on long context tasks with lower compute requirements [18]. The original SSM formulation consists of four matrices that act as gates across a continuous data stream.

$$h_{t+1} = Ah_t + Bx_{t+1} \tag{1}$$

$$y_{t+1} = Ch_{t+1} + Dx_{t+1} \tag{2}$$

In the discrete-time formulation, these matrices are discretized[1] [36] with a $\Delta$ value representing a step size across a continuous sequence.

$$\bar{A} = \exp(\Delta A) \tag{3}$$

$$\bar{B} = (\Delta A)^{-1}(\exp(\Delta A) - I)\Delta B \tag{4}$$

The original SSM formulation is linear time-invariant, allowing it to be computed as an efficient 1-dimensional convolution over a sequence. However, the Mamba SSM variant makes the B, C, and D matrices input-dependent, making them more adaptable using gating (The A matrix is determined using the HiPPO matrix formulation for long context data

---

[1]Recent work has shown that using the fixed HiPPO matrix and discretization cannot perform well in state-tracking tasks [33]. We acknowledge this approach, but we use the original Mamba implementation due to its memory-efficient selective scan kernel

storage [17]). Although this model is no longer time-invariant, it does not use activation functions, allowing the model to be computed in an $O(N)$ associative scan [5] using a parallelized, hardware-aware kernel [9].

## C. Axial Attention

Previous MSA-based models [25] [39] applied axial attention to establish relations across the sequence and MSA dimensions. Axial attention applies the attention process across items in a 2D matrix that share the same coordinates, allowing relevant row and column can be incorporated into the attention formulation [21] (Figure 4).

Axial attention has shown high performance in protein models [25]. In this data modality, sequences reach a realistic maximum length of 2000 amino acids. However, the sizes of genes are much larger than that of proteins (see III-B2).

## III. METHODS

This section provides an overview of the MSAMamba architecture, which fixes the computational complexity and context-length limitations of previous DNA MSA models. This model uses Mamba's selective scan operation along the sequence dimension, which allows sequence lengths to scale with a linear computational complexity (Proof III-B3).

Unlike axial attention, analysis across the sequence and MSA dimensions are separated in MSAMamba. This separation is done to decrease the number of relations between base pairs across the MSA that must be computed per position. Instead of comparing each base pair in the MSA using a dense attention framework, the separation of row and column processing allows each base pair to embed relevant sequence and MSA-related features independently. After running a selective scan in the horizontal dimension, the model runs multi-head attention with absolute position embeddings[2] along the MSA dimension. While this process shows quadratic scaling along the MSA dimension, most DNA MSAs do not scale past 100 species (see IV-A), making MSA-related complexity scaling trivial compared to the sequence length dimension.

An MSAMamba block consists of a horizontal selective scan, an absolutely positioned vertical attention block, and a transition MLP block to encode memory [25]. There are residual connections [20] and RMSNorm [47] blocks after the selective scan and attention operations, analogous to the `Add + Norm` block used in transformer models [46].

The selective scan operation (using the Mamba implementation) across the sequence-length dimension allows the model to attend to a context length 8 times larger than previous methods. Transformer-based methods have trained on sequence lengths of 128 base pairs, but MSAMamba can attend to 1024 base pairs per sample during training, allowing it to capture long-context relationships within DNA data.

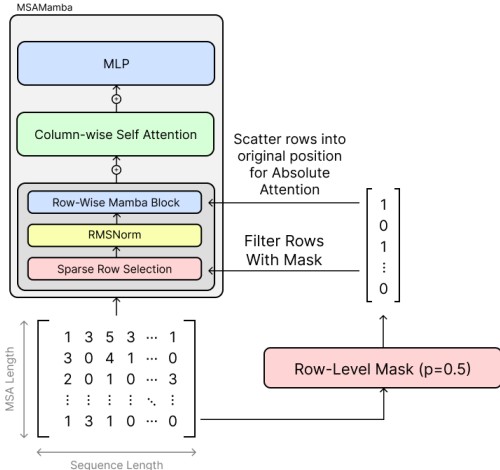

Fig. 2. Shows the row-level masking method used for sparse computations of the full MSA matrix. We mask approximately half of all additional MSA sequences based on a random probability and filter out masked rows during the selective scan process. This decreases selective scan complexity for larger batch sizes without heavily diminishing performance

## A. Row-Level Masking

To decrease the computational costs of MSAMamba further, we mask a percentage[3] of auxiliary aligned sequences in each MSA sample (Figure 2). This method allows the model to filter out masked rows during the selective scan operation, which decreases computational complexity on large batch sizes during training[4].

We compare training loss trajectories of MSAMamba with and without row-level masking to determine its effect on training performance. We find that while MSAMamba with row-level masking is slower to converge to an initial local minimum, it reaches a similar training loss level to MSAMamba without row-level masking (Figure 3).

## B. Computational Complexity Proof

In this section, we symbolically calculate the computational complexity of both axial-attention transformer models and the proposed MSAMamba architecture for DNA MSA modeling. We prove the following theorem:

$$\lim_{n \to \infty} \frac{C_{axial}}{C_{MSAMamba}} > 1 \quad (5)$$

Where $axial$ and $MSAMamba$ are both parallelized vector functions that take in an input tensor of dimension $(m, n, d)$ [5], where $n$ is the sequence length, $m$ is the number of sequences in the MSA, and $d$ is the vector function's dimension.

This theorem shows that MSAMamba's separated processing operation requires fewer computational operations than the previous state-of-the-art method (axial attention).

---

[2]Used over rotary position embeddings because the absolute position of keys is required to identify which auxiliary sequence the model is analyzing
[3]We found that a 50% masking rate was optimal for row-level masking
[4]drops overall selective scan batch size due to MSA length scaling in batch (batch size * MSA length)
[5]excludes batch size for calculation

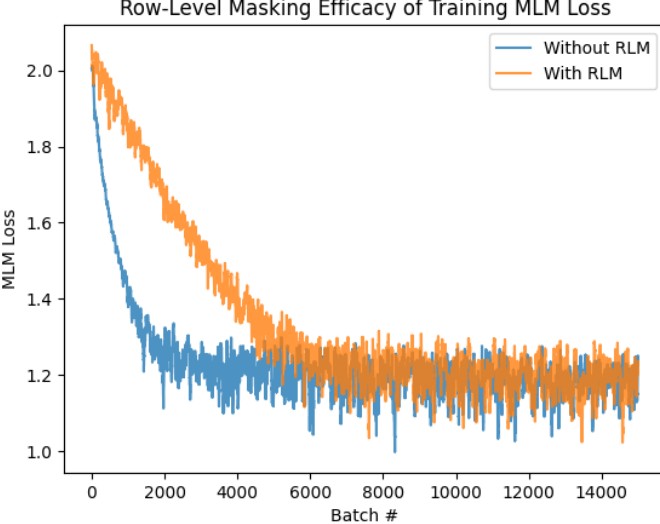

Fig. 3. Shows the efficacy of row-level masking on training masked language modeling loss across the first 15000 physical batches (batch size of 2, excluding gradient accumulation). This analysis was done on an MSAMamba model with a model dimension of 128 and a sequence length of 1024

The values, $C_{axial}$ and $C_{MSAMamba}$, represent the computational complexity of the respective models, given the baseline that one vector dot product or vector elementwise operation equates to one complexity unit.[6]

*1) Assumptions:* Some assumptions we make during the proof are as follows:

- We define $C$ as the symbolic computational complexity, which we measure in units of # of operations. An operation can denote an elementwise vector operation or a vector dot product. A matrix multiplication $\mathbb{R}^{m \times n} \cdot \mathbb{R}^{n \times d} = \mathbb{R}^{m \times d}$ is considered to be $m \times d$ total operations
- We exclude commonalities among the models (MLPs, normalization, residuals) from the complexity calculation and only include calculations that involve modeling relationships across MSA sequences
- the model size $d$ is chosen to be 1 for the sake of symbolic simplicity throughout the proof. This does not affect the output of the limit, as it is determined by the sequence length variable $n$

*2) Complexity of Axial Attention:* Axial Attention (Figure 4) involves comparing each element within the input tensor with other elements on the same $n$ and $m$ axes. Each relationship comparison involves two dot products: one during the multiplication of $Q$ and $K$ matrices and another during multiplication by the $V$ matrix. Each attention computation also consists of a softmax operation and an elementwise multiplication (scaling). The complexity calculation of axial attention for a single element in the tensor is as follows:

[6]Complexity is defined as computational complexity but is calculated similarly to time complexity. However, we do not use the term "time complexity" due to parallelization that occurs on GPUs and other AI processing units

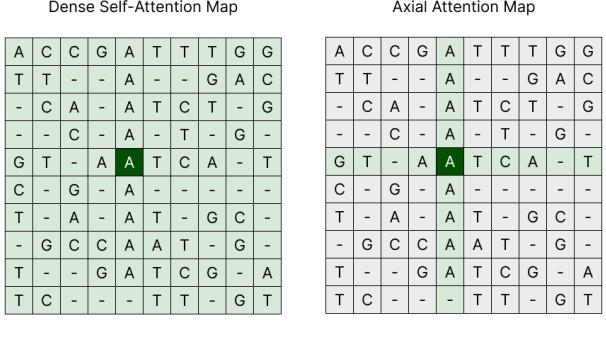

Fig. 4. A visualization of axial attention compared to fully dense attention. Axial attention significantly decreases the amount of relationships required per attention process.

$$C_{i,j} = 4(n + m - 1) \tag{6}$$

This number of dot products is computed for every element in the MSA, leading to $mn$ axial attention computations. The computational complexity of axial attention can be calculated with this information:

$$C_{axial} = 4nm(n + m - 1) = 4n^2m + 4nm^2 - 4nm \tag{7}$$

*3) Complexity of MSAMamba:* MSAMamba leverages the mamba operator for every sequence in the MSA, while the number of attention processes scales linearly based on sequence size.

Mamba's Selective Scan involves 3 vector dot products[7] and four elementwise multiplications[8] per sequence. We use this information to calculate Mamba's computational complexity:

$$C_{mamba} = n(3 + 4) = 7n \tag{8}$$

The column-wise attention process involves comparing every element of the same base pair index across MSA's, which leads to $m^2$ total comparisons per item in the sequence. Each relation consists of 2 dot products, as described in the axial attention complexity analysis in III-B2. This leads to a complexity of

$$C_{attention} = 2nm^2 \tag{9}$$

The overall computational complexity of MSAMamba is

$$C_{MSAMamba} = 7n + 2nm^2 \tag{10}$$

[7]Creation of B, C, and D matrices. Assumes 1:1 scale from input to inner SSM dimension
[8]A, B, C, and D gating matrices

*4) Confirming MSAMamba's Lower Time Complexity:* We evaluate the limit defined in Eq. 5 given the time complexities calculated in Eq. 7 and 10. This gives the equation:

$$\lim_{n \to \infty} \frac{4n^2m + 4nm^2 - 4nm}{7n + 2nm^2} > 1 \qquad (11)$$

To evaluate the limit to infinity, we take the terms in the numerator and denominator with the highest degree[9] of $n$, leading to the equation:

$$\lim_{n \to \infty} \frac{2n^2m}{n(2m^2 + 7)} > 1 \qquad (12)$$

Since the degree of $n$ in the numerator is higher than the degree of $n$ in the denominator, we can ignore constant term coefficients and prove the following:

$$\lim_{n \to \infty} \frac{n^2}{n} > 1 \qquad (13)$$

$$\lim_{n \to \infty} n > 1 \qquad (14)$$

Therefore, the computational complexity (based on the number of vector calculations) of axial attention-based DNA MSA models increases by an order of magnitude faster than MSAMamba when scaling sequence length.

*5) Summary of Proof and Relation To Proposed Model:* The above proof shows that MSAMamba is more computationally efficient (concerning the number of vector calculations) at larger context lengths. DNA sequences consist of genes that can be up to millions of base pairs long and genomes made of billions of base pairs. MSAMamba's scaling properties show they can model longer DNA sequences more efficiently than current axial attention-based implementations.

## IV. Datasets

This section gives an overview of the datasets used to pre-train and fine-tune MSAMamba, why they were selected for training, and data preprocessing for training tasks.

### A. Pre-Training: MultiZ100Way

During model pre-training, we leverage the MultiZ100Way dataset, which consists of an MSA of the length of the human genome without any gap sequences[10] in the human sequence. It also consists of 99 auxiliary aligned sequences (with gap sequences) from related species. This data has been curated from the public UCSC Genome Browser [34]. We use a modified version of this dataset, which excludes ten auxiliary sequences of organisms that are very similar to those of humans [4]. This modification was done to decrease training time and memory requirements while losing minimal auxiliary information.[11]

---

[9]If two terms with the same degree are present, we take the one with the highest coefficient assuming $m = 90$

[10]Gap sequences occur in MSAs when alignment moves around nucleotides to fit the proper evolutionary configuration, leaving placeholders for locations affected by shift/insertion/deletion mutations

[11]The MultiZ90Way is publicly accessible through HuggingFace datasets [28]

---

**Algorithm 1** MSAMamba Masked Language Modeling
| |
| --- |
| **Input:** MSA $x$ : (B, M, L, D), $M_{row}$ : (B, M), $y_t$ : (B, L, D), lr, $\theta$ (Model Params) |
| **Output:** $y$ : (B, L, D) |
| $h_0$ = mask($x$, p=0.15) |
| **for** $i = 1$ **to** $n_{layers}$ **do** |
| $\quad h_{sparse}$ = $h_i[M_{row}]$ |
| $\quad O_{mamba}$ = scatter(Mamba($x_{sparse}$), $M_{row}$) + $h_i$ |
| $\quad O_{att}$ = SelfAttention($O_{mamba}$) + $O_{mamba}$ |
| $\quad h_{i+1}$ = MLP($O_{att}$) |
| **end for** |
| loss = CrossEntropy($h_{n_{layers}-1}[h_0 = MASK]$, $y_t$) |
| $\theta \leftarrow$ AdamW(lr) |

This dataset was used to train MSAMamba and all MSA-based baseline models[12]. The same random seeds were also used for data shuffling and batch loading during pre-training for MSAMamba and other baseline models.

*1) Data Preprocessing:* The initial training data was collected from the MultiZ100Way dataset by sampling random locations across the genome and selecting DNA sequences based on the required context length for training (128, 512, or 1024).[13]

Data in the MultiZ100Way dataset was parsed using a tokenizer with a vocabulary size of 6. This consists of 4 nucleotides, one token for gap sequences, and one mask token. There was no need for `<PAD>` tokens due to all excerpts from the dataset being the same length.

This data was preprocessed based on the masked language modeling algorithm. This involves masking 15% of the sequence, where 80% of masked tokens are replaced with the `<MASK>` token, 10% is replaced with a random token, and the final 10% is not replaced [11].

***Note:*** *Only the top sequence in the MSA (the human sequence) is masked due to the focus on the human genome, with other genomes being additional information*

### B. Evaluation

*1) Variant Effect Prediction Tasks:* We use the OMIM and ClinVar Datasets during the evaluation process. The OMIM dataset relates gene sequences to different genetic disorders and their forms [19], while ClinVar relates aggregated gene variance information to overall human health [27]. Fine-tuning on this dataset evaluates a DNA MSA model's ability to perceive overall and individual gene relationships to determine its properties. The addition of MSA information provides key evolutionary information that is useful for these tasks [4].

These two datasets were used at three sequence lengths: 128, 512, and 1024. Previous DNA-MSA transformer models were trained on a sequence length of 128 [4]. However, MSAMamba is trained on sequence lengths of 128, 512,

---

[12]Non-MSA models used as baselines were trained on the regular human genome without MSA augmentation

[13]We were unable to train on the entire genome due to lack of computational power

and 1024. We compare evaluations from the fine-tuning processes across these increasing context windows to determine MSAMamba's relative efficacy when parsing longer MSA sequences.

The original dataset consisted of 128-length sequences. We modified these original sequences to include the area around the original sequence to add up to larger context lengths. This tests models' abilities to analyze specific mutations and segments within longer sequences.

All sequences were retrieved from the MultiZ90Way database given each sequence's chromosome index, start indices, and end indices. These sequences were not masked but passed as a tuple with a binary label as the fine-tuning target.

*2) Genomic Benchmark Tasks:* MSAMamba and other relevant models were also evaluated on the GenomicBenchmarks dataset [15]. This dataset consists of 8 different tasks relating to sequence-level classification. The original GenomicBenchmarks datasets are single-sequence, containing only the human genome. However, we use start indices, stop indices, and chromosome metadata from the datasets along with the MultiZ90Way database to generate MSA versions of these evaluation datasets.

These datasets were not modified for different sequence lengths and were only trained on their original sequence lengths.

***Note:*** *Ethical considerations were carefully addressed during the data curation/processing step. All genome data used in this study were obtained and modified from publicly available datasets (e.g., MultiZ100Way, OMIM, ClinVar)*

## V. TRAINING

This section gives an overview of the different methodologies and hyperparameters used during the training process. We also provide different model sizes and configurations tested during the process.

Four MSAMamba models were trained to determine the architecture's efficacy (see Table I). Three models were trained on DNA sequence lengths of 128, 512, and 1024, respectively (with row-level masking). The fourth model was trained on a sequence length of 1024 without row-level masking to determine its effect on training performance (see III-A). MSAMamba was trained on batch sizes that amounted to a total of 49152 nucleotides per logical batch (excluding augmented MSA sequences).[14]

The masked language modeling task (Algorithm 1) was used for pre-training, with 15% of each sequence being masked [11]. Both models were trained on the MultiZ90[15] genome dataset (see IV-A)

### A. Baseline Models

The primary baseline model we compare to is GPN-MSA, an axial-attention-based DNA modeling architecture that was

---

TABLE I

TABLE OF MODEL CONFIGURATIONS THAT UNDERWENT THE TRAINING, FINE-TUNING, AND EVALUATION PROCESSES WITH COMPARISON TO BASELINE MODELS WITH SIMILAR PARAMETERS

| $d_{model}$ | $d_{ssm}$ | $n_{layers}$ | SEQ. LEN | ROW SPARSE |
|---|---|---|---|---|
| 128 | 256 | 3 | 128 | $\checkmark$ |
| 128 | 256 | 3 | 512 | $\checkmark$ |
| 128 | 256 | 3 | 1024 | $\checkmark$ |
| 128 | 256 | 3 | 1024 | $\times$ |

trained on multiple sequence alignments of size 128. We evaluated the original pre-trained GPN-MSA model on sequences of 128, 512, and 1024 base pairs to compare to MSAMamba at respective sequence lengths. The model had a dimension of 256 and consisted of 6 transformer layers. We trained the model with hyperparameters provided in the original paper.[16].

In addition, we use benchmarks from CADD, PhyloP, and phastCons in DNA variant effect prediction. Results for these models on 128 sequence length inputs were used from baseline metrics in GPN-MSA's evaluations. We evaluate these models on sequence lengths of 512 and 1024 on the same dataset used in evaluating MSAMamba. These models were fine-tuned for the given task based on the default provided hyperparameters and configuration [34]. We also evalute HyenaDNA, DNABERT, and a CNN classifier as baseline models on Genomic Benchmarks tasks.

### B. MSA Mamba Training

When training MSAMamba, we swept across different magnitudes of learning rates and weight decays. We also tested with two primary configurations of betas in the AdamW optimizer, and we experimented with warm-up [14] and cosine annealing learning rate [32] schedulers.

*1) Model Sizes:* We trained MSAMamba on a size of 3 total layers with a model dimension of 128. The SSM layer's dimension was scaled up by two times the model dimension, and the transition MLP module's magnification rate was 4x (similar to that of transformers). Model depth was kept constant to prevent external factors from influencing the model's long-context modeling performance measurements.

### C. Optimizers and Schedulers

We used the AdamW optimizer[17] during the pre-training and fine-tuning processes. We also used a warm-up scheduler for the first 10% of gradient steps. A weight decay of `1e-3` was used throughout pre-training and fine-tuning. For both tasks, we used betas of `(0.9, 0.95)`[18].

---

[14]batch size 48 for 1024 sequence length, batch size 96 for 512 sequence length, batch size 384 for 128 sequence length

[15]Modified from MultiZ100Way to exclude the ten genomes most similar to humans [4]

[16]learning rate: `1e-4`, weight decay: 0.01, 30K batches with warm-up scheduler for first 1K batches

[17]We also tested the SGD optimizer due to initial issues in the adaptive training algorithm leveraged by Mamba. However, we found minimal difference between the two

[18]We experimented with a second beta of 0.99, but we discovered that it would lead to slower convergence and moved it to 0.95

## D. Hyperparameter Selection

We used a learning rate of `3e-5` for pre-training across all context lengths and row-level masking configurations. For fine-tuning, we used a learning rate of `3e-4`. We swept across the following learning rates during the pre-training process: `8e-3, 2e-3, 3e-4, 3e-5, 8e-6`, and found that `3e-5` was the highest performing learning rate in all model configurations.

Due to limited resources, the model was trained on a total sequence length of 2048 base pairs per physical batch. To compensate, we use a gradient accumulation across batches. This led to 49,152 base pairs in the provided MSA input per theoretical batch[19]. Validation loss was calculated after every two gradient updates. Half-size batch sizes were used when fine-tuning the model.

## E. MSAMamba Evaluation

The base MSAMamba model was modified during the evaluation process for sequence classification tasks. This was done by appending a pooler module that takes the last hidden state of each sequence in the batch and passes it as input to a single linear layer. A classifier linear layer follows this. Dropout was placed in between these layers with `p=0.25`. The final classifier layer was followed by a sigmoid function, which was used to compute binary cross-entropy loss as the objective function for genomic benchmarks and variant effect prediction tasks.

## VI. RESULTS

We evaluate MSAMamba on the OMIM and ClinVar datasets for variant effect prediction on missense mutations. This task tests the ability to leverage MSA information, as mutation predictions rely heavily on evolutionary data provided in aligned sequences [4]. This task was chosen to compare to current SOTA[20] MSA and non-MSA DNA models with similar training data. We assess the chosen models at increasing context lengths[21] to demonstrate MSAMamba's improved prediction capabilities at larger context lengths. General DNA models (MSAMamba, GPN-MSA) were fine-tuned using pooler and classification layers (see IV-B1), while task-specific methods (PhastCons, PhyloP) were used without fine-tuning. At each context length threshold, we evaluate the MSAMamba model trained at the respective sequence length. We evaluate baseline models based on released pre-trained models of similar size to MSAMamba's model dimensions (see Table I).

Results (Figure 5) show that while GPN-MSA's performance on OMIM and ClinVar variant effect prediction decreases with increasing context length, the performance of MSAMamba increases. This is likely due to GPN-MSA's difficulty in analyzing global relationships across longer sequences. MSAMamba shows the most significant performance

increase across context length and exceeds SOTA DNA MSA models by a margin of $\approx 0.2$ AUROC/AUPRC at a context length of 1024.

In addition, we evaluate MSAMamba on the GenomicBenchmarks datasets [15] (modified using methods in IV-B2). Evaluations show that MSAMamba performs better than alternative models in 2 of 8 tasks (Table V-E). These tasks are based on evolutionary relationships across species and require attention to global relationships. MSAMamba's longer context training and MSA data augmentation provide an advantage in these features. MSAMamba shows minor performance differences from the state-of-the-art in other GenomicBenchmarks tasks (maximum 2.4%). HyenaDNA shows high performance in 2 tasks due to its training on $2^{20}$ base pairs per batch, making it highly attuned to global DNA relationships [35].

## VII. DISCUSSION

### A. Summary

By incorporating a subquadratic selective scan operation and separating processing along the sequence and MSA dimensions, MSAMamba achieves efficient and scalable inference on long DNA sequences. Our experiments demonstrate that MSAMamba exceeds the performance of state-of-the-art MSA and single-sequence models in four GenomicBenchmarks tasks (Table V-E). In addition, the model shows performance exceeding current state-of-the-art DNA MSA models in long-context variant effect prediction (Figure 5). The row-sparse method used in MSAMamba's training process further enhances computational efficiency during the training process (Figure 3), making MSAMamba a viable and powerful tool for large-scale DNA analysis.

### B. Analysis of Results

In this section, we provide a comprehensive analysis of the results on fine-tuning benchmarks that MSAMamba received.

*1) Variant Effect Prediction:* We evaluated baseline models and the proposed MSAMamba model on the Variant Effect Prediction task using the OMIM and ClinVar datasets. We used a different set of baseline models that were proposed for variant effect prediction.[22]

- phastCons - a hidden Markov model (HMM) for identifying conserved elements within a DNA multiple sequence alignment. This model predicts a nucleotide-level score of conservation, which can be used to determine DNA variants/mutations [44]
- PhyloP - similar to phastCons, but it computes p-value probabilities per nucleotide for evolutionary conservation [38]
- CADD - an SVM-based model to predict mutation sites [42]

***Note on data***: *the ClinVar and OMIM datasets are traditionally single sequence models, so we extract MSA versions*

---

[19]physical batch size $\times$ gradient accumulation iterations

[20]GPN-MSA [4] is the current state of the art for MSA-based processing. PhastCons, PhyloP, and CADD [42] are also evaluated

[21]sequence lengths of size 128, 512, and 1024

[22]Other models were proposed specifically for DNA language modeling and do not have the same level of inductive bias integrated into variant effect prediction-aligned models

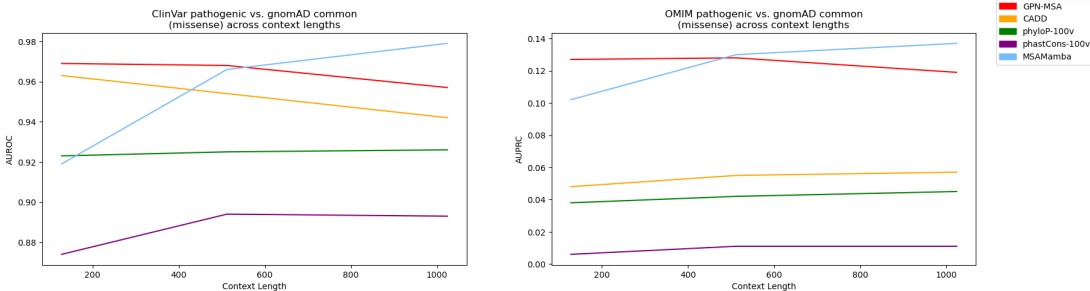

Fig. 5. Graphs of MSAMamba (with row-level masking) and related models' performance on OMIM (AUPRC) and ClinVar (AUROC) missense mutation detection. The $x$ axis shows the context length of the evaluated sequences

| Task Name | CNN | DNABERT | HyenaDNA | GPN-MSA | MSAMamba |
|---|---|---|---|---|---|
| Mouse Enhancers | 69.0 | 66.9 | **85.1** | 76.4 | 82.7 |
| Coding vs Intergenomic | 87.6 | **92.5** | 91.3 | 90.3 | 90.0 |
| Human vs Worm | 93.0 | 96.5 | 96.6 | **98.9** | 98.5 |
| Human Enhancers Cohn | 69.5 | 74.0 | **74.2** | 73.1 | 72.7 |
| Human Enhancers Ensembl | 68.9 | 85.7 | 89.2 | **89.3** | 88.8 |
| Human Regulatory | 93.3 | 88.1 | 93.8 | 93.5 | **94.4** |
| Human Nontata Promoters | 84.6 | 85.6 | **96.6** | 90.9 | 94.2 |
| Human OCR Ensembl | 68.0 | 75.1 | 80.9 | 76.8 | **82.5** |

TABLE II
EVALUATION OF MSAMAMBA (WITH ROW-LEVEL MASKING), GPN-MSA, AND OTHER SINGLE SEQUENCE MODELS ON GENOMICBENCHMARKS TASKS USING TOP-1 ACCURACY (%) METRIC

*of the data from the UCSC Genome Browser [34] given the start and stop coordinates of each input sequence*

The following models are traditional mutation/variant effect prediction tools, and they are used for evaluation to compare to results found in GPN-MSA, which leveraged the same models for evaluation.

When evaluating the models on both datasets at a context length of 128, MSAMamba showed performance worse than GPN-MSA, as well as worse performance compared to CADD in ClinVar evaluations. This occurs due to GPN-MSA and other algorithms' affinity to shorter DNA sequences. Since GPN-MSA uses the fully connected mixer methodology [23] of self-attention, it can understand deeper relationships at smaller context lengths. In contrast, MSAMamba is akin to a semiseparable matrix mixer, which lacks the level of relationship establishment that a transformer has.

However, MSAMamba shows equivalent performance compared to GPN-MSA at a context length of 512 and improves on GPN-MSA's performance at a context length of 1024. This occurs due to MSAMamba's specific focus on longer sequences during training. In contrast, GPN-MSA and other methods were trained on 128-length DNA sequences. MSAMamba's long-context representation abilities outperform GPN-MSA's high-resolution relationship evaluation through attention. In addition, GPN-MSA embeds a full MSA column as a single token. MSAMamba's more fine-grained approach with column-wise attention provides more inductive bias for

extracting MSA-related features (coevolution, conservation), which can improve performance when evaluating longer context relationships [1].

*2) Genomic Benchmarks:* We evaluate MSAMamba on the genomic benchmarks dataset, which consists of 8 separate tasks. The "Mouse Enhancers" task is a dummy task with a small dataset, used for testing the fine-tuning process. Both "Coding vs Intergenomic" and "Human vs Worm" datasets are demo datasets with medium-size data. All other tasks are full-size datasets with consistent reproducible results.

*Note on data: the genomic benchmarks datasets are traditionally single sequence models, so we extract MSA versions of the data from the UCSC Genome Browser [34] given the start and stop coordinates of each input sequence*

We evaluate MSAMamba along with the following baseline models for comparison:

- a CNN architecture with one-dimensional sets of short convolutions, along with standard ReLU, BatchNorm, and MaxPooling layers
- DNABERT (110 million parameters) - a BERT transformer architecture trained to represent DNA sequences
- HyenaDNA - a long convolution-based subquadratic architecture for DNA processing. The HyenaDNA-tiny version was used with a model dimension of 128 and a sequence length of 16k
- GPN-MSA - a transformer model that processes DNA MSAs (note: all other models work on single sequence

only). We used a model with a dimension of 128 and used a checkpoint trained on sequences of length 128[23]

In all 5 relevant tasks, scores between HyenaDNA, GPN-MSA, and MSAMamba are within 2% of each other. MSAMamba shows the best performance in the "Human Regulatory" and "Human OCR Ensembl," which are both tasks related to non-regulatory open-chromatin regions (OCR)[24]. MSAMamba's training data, while randomly sampled from the human genome, has a bias towards open chromatin regions, leading to their improved performance on these tasks.

HyenaDNA shows the best performance on regulatory region analysis (enhancers and promoters). Since the HyenaDNA model that was used was trained on sequence lengths of 16k, it can better understand long-context regulatory relationships.

*3) Conclusions:* Overall, an analysis of benchmarks shows that MSAMamba shows state-of-the-art performance in evaluating and classifying open chromatin regions, while previous work shows better performance on analyzing regulatory sequences. MSAMamba also shows high performance in long-context variant effect prediction tasks.

## C. Applications

With high performance in open chromatin region representation and long-context variant effect prediction, MSAMamba can be applied to genomic tasks that require an evolutionary basis of understanding. This is useful in mutation detection tasks and detecting conserved or coevolved locations within a DNA sequence. In addition, DNA language models can be combined with protein models to assist in drug discovery/protein design with genomic priors.

## D. Limitations

While MSAMamba provides benefits in computational complexity and context length scaling. However, it loses the expressiveness that full self-attention inherently possesses. In the perspective of matrix mixers [23], full self attention consists of a full matrix. In contrast, Mamba's mixer matrix is semiseparable, which leads the MSAMamba model to be inherently causal [10]. Building this model with a bidirectional version of Mamba may be more performant [41] [23].

Many protein model architectures have inductive biases that improve performance. However, current DNA language models do not have this specificity. Specific blocks that inherently compute common DNA features, such as coevolved/conserved positions [13] and short motifs [3], may improve performance DNA language modeling performance even further.

## E. Future Work

In future studies, we hope to address the following:

- Testing MSAMamba with a bidirectional subquadratic model

---

[23]Trained on smaller length due to this value being used in the original paper and computational constraints

[24]While the "Human Regulatory" task contains both regulatory and OCR regions, it contains a majority of OCR regions

- Creating an operator that is attuned to extracting MSA-level features, such as coevolved sequences and motifs

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
