# OpenReview forum: "MSAMamba: Adapting Subquadratic Sequence Models To Long-Context DNA MSA Analysis"
_IEEE.org/ICIST/2024/Conference — IEEE ICIST 2024 Conference Submission_

### Official Review · Reviewer_6Hcf · 2024-08-21
**This manuscript has a certain degree of innovation and clear simulation figures. It is recommended to accept this paper for publication in IEEE ICIST 2024.**

**Rating:** 7
**Confidence:** 4

**Review:**

This manuscript has a certain degree of innovation and clear simulation figures. Please answer the following review questions.

The paper mentions that MSAMamba leverages a selective scan operation along the sequence dimension to enhance efficiency. Could you provide more details on how this selective scan operation works? How does it differ from the standard self-attention mechanism used in traditional transformers?

The paper introduces a row-sparse training method to reduce the computational overhead of the selective scan operation during model training. Could you provide more details on how this row-sparse training method works and how it was implemented? What are the performance gains observed as a result of this method?

---

### Official Review · Reviewer_8PFU · 2024-08-21
**The paper is logically clear, the simulation results are convincing, and it is recommended for publication.**

**Rating:** 7
**Confidence:** 3

**Review:**

In this paper, by innovatively using selective scanning operations and separating the processing of sequence length and MSA dimension, the proposed architecture effectively reduces the secondary complexity associated with self-attention on a large scale. This method not only improves efficiency, but also respects the inductive bias of MSA levels, which is crucial for accurate genomic analysis. The reviewer has the following questions to discuss with the authors:

1. How exactly does the selective scan operation work along the sequence dimension, and what makes it more efficient than the traditional self-attention mechanism used in transformers?

2. What challenges were encountered in separating the processing of sequence length and MSA dimensions, and how does this separation contribute to the overall efficiency and performance of MSAMamba?

3. Can you explain in more detail how the row-sparse training method reduces the computational overhead of the selective scan operation? Are there any trade-offs associated with this method?

---

### Official Review · Reviewer_4WJW · 2024-08-22
**This paper is innovative and has made a certain contribution to genome analysis. It is recommended to  publish.**

**Rating:** 7
**Confidence:** 4

**Review:**

This paper is innovative and has made a certain contribution to genome analysis. There are the following issues that need to be discussed：
1. The authors mentioned that  MSAMamba can increase the training context length of previous methods by 8x, has this data result been validated in this paper? If yes, please provide more details.
2.  A row-sparse training method is developed to reduce the computational overhead of the selective scan operation. Please provide a theoretical analysis of how this method reduces computational overhead and the advantages of using it in this paper.

---

### Decision · Program_Chairs · 2024-09-08

Accept (Oral)